# Adapting the Motivated Strategies for Learning Questionnaire to the Japanese Problem-Based Learning Context: A Validation Study

**DOI:** 10.3390/children10010154

**Published:** 2023-01-12

**Authors:** Osamu Nomura, Yuki Soma, Hiroshi Kijima, Yasushi Matsuyama

**Affiliations:** 1Department of Health Sciences Education, Hirosaki University, Hirosaki 036-8562, Japan; 2Centre for Community-Based Health Professions Education, Hirosaki University, Hirosaki 036-8562, Japan; 3Faculty of Education, Hirosaki University, Hirosaki 036-8560, Japan; 4Medical Education Center, Jichi Medical University, Shimotsuke 329-0498, Japan

**Keywords:** self-regulated learning, validation, measurement, problem-based learning, motivated strategies for learning questionnaire

## Abstract

The COVID-19 pandemic has greatly changed medical education, and medical trainees’ self-regulation has become more emphasized. In Japan, the concept of self-regulated learning has not been fully applied in health profession education due to a lack of effective measurement tools. We aimed to validate the translated Japanese version of the Motivated Strategies for Learning Questionnaire in the context of Problem-Based Learning (J-MSLQ-PBL). The questionnaire employs a seven-point Likert-type scale with 81 items and is categorized into two sections: motivation and learning strategies. An exploratory factor analysis (EFA) was conducted by using Promax rotation to examine the factor structure of the scale, using the collected data from 112 Japanese medical students. Factor extraction was based on a scree plot investigation, and an item was accepted when the factor loading was ≥0.40. In the motivation section, the extracted factors from the EFA were well aligned with the subscales of the original MSLQ, including “Self-Efficacy for Learning and Performance”, “Task Value”, “Self-Efficacy for Learning and Performance”, “Test Anxiety”, “Extrinsic Goal Orientation”, and “Intrinsic Goal Orientation”. In the learning strategies, the extracted factors poorly matched the structure of the original subscales. This discrepancy could be explained by insufficient translation, the limited sample size from a single medical school, or cross-cultural differences in learning strategies between Western and Japanese medical students. Only the motivation part of the J-MSLQ-PBL should be implemented to measure the competency elements of self-regulated learning in Japan.

## 1. Introduction

COVID-19 has substantially changed medical education. Social distancing is strongly recommended during a pandemic, and in-person educational activities in medical schools and hospitals have been suspended; therefore, the modality of learning for medical students has changed to an online mode [1,2]. This results in fewer opportunities for trainees in medicine to meet their classmates and talk with the faculty and mentors at medical schools [2]. With the increased use of online learning during the pandemic, students spent less time in live learning opportunities and more time in self-directed learning activities.

Medical students’ autonomy and self-regulation should be more emphasized during and post-COVID-19 pandemic, as self-regulated learning is a fundamental process that allows students to adapt to the unusual situation (e.g., pandemic) and to develop planning, prevision, and monitoring of their learning activities and wellness [3].

Self-regulated learning, defined as learners’ active participation in the learning process from metacognitive, motivational, and behavioral perspectives, is becoming even more prominent in the new normal era [4,5,6,7]. Zimmerman’s self-regulated learning theory suggests how motivations work in a cyclic process in three phases: forethought, performance, and self-reflection [8]. In the first phase of forethought, learners set goals and choose strategies to achieve them by utilizing their motivational beliefs, such as self-efficacy, values, and interests. Next, learners observe and control themselves in the performance phase to attain their goals. In the last phase of self-reflection, individuals reflect on their previous performance to prepare for new goals for future learning (i.e., the new foresight phase) [9].

The Motivated Strategies for Learning Questionnaire (MSLQ) is one of the most widely used measurements designed to assess the competency elements in self-regulated learning in pedagogy. In addition, this measurement is widely used in health professions’ education research. Cook et al. aimed to validate the MSLQ in medical trainees and showed that several factors of their MSLQ data demonstrated a similar psychometric profile to that of original scales studied in educational psychology [10]. Another study investigated the medical students’ changes in the self-regulated learning process during the transition to clinical learning in the first clinical year in Australia [11].

However, in Japan, self-regulated learning has yet to be fully applied in medical education due to a lack of effective measurement tools [12]. Problem-based learning (PBL) is one of the ordinary teaching strategies that facilitates students’ self-regulated learning competencies in Japanese undergraduate medical education. This study, therefore, aimed to collect and examine validity evidence for the Japanese version of the MSLQ adapted to the PBL context.

## 2. Materials and Methods

### 2.1. Design and Participants

This was a secondary analysis study using a database, which was initially collected to examine the impact of self-regulated learning during the problem-based learning (PBL) course [11] at Jichi Medical University, Japan, on medical students’ professional identity formation. Of all 124 third-year medical students invited to participate in this study, 112 agreed.

### 2.2. Measurements

The MSLQ, which is a seven-point Likert-scale survey, includes two sections: motivation and learning strategies [12]. The motivation section includes 31 items assessing three domains: goal orientation, self-belief, and test anxiety. The learning-strategies section includes 50 items assessing three domains: the use of cognitive strategies, metacognitive strategies, and resource management. The principal investigator of the research project (Y.M.) and the supervisor (A.L.J.) created a Japanese version of the MSLQ by translating all 81 items into Japanese, with backtranslation [13]. In the translation process, we adapted the item descriptions to the context of PBL to make the MSLQ suitable for assessing the participants’ self-regulated learning competencies during the PBL course. We named this scale the J-MSLQ-PBL.

### 2.3. Context

The one-day PBL program for third-year medical students was divided into four segments: (1) an opening case discussion for the formulation of the self-study objectives, (2) a self-study period for objectives and preparation for subsequent group discussion, (3) a group discussion that included within-group information sharing, and (4) a 60 min wrap-up lecture from a specialist. This survey was conducted in the orientation phase of the PBL course.

### 2.4. The Theoretical Framework of Validation

According to Kane [14], the validation of a measurement method requires gathering evidence to examine the four key inferences: (1) the scoring of a single observation (scoring), (2) using the primary observation score to generate the whole test performance (generalization), (3) inferring the real-life performance from the test performance (extrapolation), and (4) interpreting this information to make a decision (implication). In addition, recent validation studies in health sciences education have commonly used Kane’s framework for translating psychomimetic tools in English into other languages [15,16].

### 2.5. Analysis

We conducted an exploratory factor analysis (EFA) with the maximum likelihood method and Promax rotation to examine the factor structure of the Motivation and Learning Strategies Scales of the J-MSLQ. Factor extraction was based on parallel analysis, and an item was accepted when the factor loading was ≥0.40. Due to the small sample size, a confirmatory factor analysis was not conducted. The Kaiser–Meyer–Olkin (KMO) test was used to test the suitability of the scale for the sampling adequacy. Cronbach’s alpha was calculated as a measure of internal reliability. All data analyses were conducted in R (Version 4.2.1) and R studio (2022.07.2 Build 576) with the packages psych (version 2.2.9) and GPArotation (version 2022.10-2).

### 2.6. Ethics

This study was approved by the Jichi Medical University Clinical Research Ethics Committee (reference number: 18–168). Informed consent was obtained from all participants.

## 3. Results

Table 1 and Table 2 show the mean score, standard deviation, median, and first and third quartiles in items of the J-MSLQ-PBL, respectively. Q5, Q6, Q15, and Q31, which were categorized as “Self-Efficacy for Learning and Performance”, tended to have low scores (less than three points). In contrast, Q4, Q17, and Q23, which were categorized as “Task Value”, tended to have high scores (more than five points). In the items of the Learning Strategies Scale, trends in scores were not observed.

### 3.1. Factor Analysis

Because of the negative correlation with the total scale, Q3 of the Motivation Scale was excluded from the EFA. The KMO tests for the Motivation and Learning Strategies Scales were 0.788 and 0.754, respectively. Based on parallel analysis, the 30-item Motivation Scale and the 50-item Learning Strategies Scale suggested six and five factors, respectively. The factor loadings and proportions of variance explained by the factors are outlined in Table 3 and Table 4.

In the Motivation Scale (Figure 1), the first factor, which explained 12.2% of the total variance in the data, was labeled “Self-Efficacy for Learning and Performance” based on the high loadings of Q6, Q15, Q20, and Q31. The second factor, which explained 9.9%, was labeled “Task Value” based on the high loadings of Q17, Q23, Q26, and Q27. The third factor, which explained 8.5%, was labeled “Control of Learning Beliefs and Self-Efficacy for Learning and Performance” based on the high loadings of Q2, Q5, Q18, Q21, and Q29. The fourth factor, which explained 8.0%, was labeled “Extrinsic Goal Orientation” based on the high loadings of Q7, Q11, Q13, Q22, and Q30. The fifth factor, which explained 7.1%, was labeled “Test Anxiety” based on the high loadings of Q8, Q14, Q19, and Q28. The last factor, which explained 6.7%, was labeled “Intrinsic Goal Orientation” based on the high loadings of Q1, Q10, Q24, and Q25. In total, 52.4% of the total variability in the data was explained by the factor structure.

In the Learning Strategies Scale (Figure 2), the first factor explained 10.9% of the total variance in the data. The second to fifth factors explained 8.9%, 8.5%, 6.4%, and 5.2%, respectively. In total, 39.9% of the total variability in the data was explained by the factor structure. There were no trends in the categories of items with high loadings in each factor; hence, labeling factors according to loadings was difficult.

### 3.2. Internal Reliability of the Motivation Scale and Subscales

After excluding the items with a factor loading less than 0.40, the Cronbach’s alpha of the overall Motivation Scale was 0.87 (26 items). The six subscales were as follows: 0.88 (four items), 0.81 (four items), 0.80 (four items), 0.79 (five items), 0.72 (four items), and 0.64 (four items) for “Self-Efficacy for Learning and Performance”, “Task Value”, “Control of Learning Beliefs and Self-Efficacy for Learning and Performance”, “Extrinsic Goal Orientation”, “Test Anxiety”, and “Intrinsic Goal Orientation”, respectively (Table 3). The internal reliability of the Motivation Scale, as well as the subscales other than goal setting, was adequate (α > 0.70).

## 4. Discussion

This study aimed to collect and examine validity evidence for the Japanese version of the MSLQ adapted to the PBL context regarding Kane’s four steps of validity arguments [14,15]. In addition, we found that the internal structure of the motivation section of the J-MSLQ-PBL was consistent with the theory of self-regulated learning; however, the learning-strategies section did not align with the original structure of the MSLQ.

### 4.1. Scoring Inference

One of the potential reasons for the inconsistency of the learning-strategies section between the J-MSLQ-PBL and the original MSLQ could be the insufficient translation process for the J-MSLQ-PBL. While the J-MSLQ-PBL was developed using the backtranslation method, the method could be inappropriate for some translations whose topic is sensitive to sociocultural factors such as the learning culture, educational system, and differences in cultural backgrounds. This could be a barrier to correctly translating assessment tools in health-sciences education. The descriptions of the motivation-section items were concise because the section focuses on the planning of goal setting for learning at an individual level. However, the descriptions of the learning-strategies section were more complicated because these subscales refer to applying strategies, monitoring performance, and reflecting on performance in the self-regulated learning process [17]. In addition, the learners’ behavior related to the subscale on profound learning (e.g., critical and metacognitive thinking) and interpersonal learning (e.g., peer learning and help-seeking) could be susceptible to cultural influences. To deal with this issue, cross-cultural survey guidelines recommend using a team translation model that employs bilingual experts to ensure proper translation and cross-cultural and linguistic equivalences between the two language survey versions [18].

### 4.2. Generalization Inference

The internal structure of the motivation section was consistent with the original MSLQ. In addition, Cook et al. [10] reported that the motivation section of the MSLQ was well-validated in medical residents by performing a correlation, reliability, and factor analysis. Furthermore, Miyabe also conducted a validation study of the Japanese version of the MSLQ with first-year nursing students [19]. The current study furthermore examined the validity evidence of the J-MSLQ-PBL and demonstrated that the scale’s internal structure is consistent with the SRL theory. This indicates that the J-MSLQ-PBL for the PBL context can be generalizable to measure Japanese medical students’ SRL competency elements for learning in PBL. However, this study was conducted in a single private medical school in Japan; thus, further validation study in another type of institution is needed to expand the generalizability. Furthermore, a more robust statistical analysis, such as a structural equation model using a larger sample size, would improve the generalizability of the J-MSLQ-PBL.

### 4.3. Implication Inference

Our study also showed that the results of EFA for the learning-strategies section did not align with the SRL theory. There are several possible reasons. Most of Japan’s educational contents in preclinical medical education are didactic, even if PBL is partially included in the curriculum [20]. Thus, Japanese medical students have fewer opportunities to develop their skills in learning strategies than medical students in other countries [21]. In this sense, there was the possibility of a content validity issue; Japanese medical students perhaps could not understand the meaning of the descriptions in the learning-strategies sections because they are less experienced in using these skills. In addition, it has been reported that the group dynamics of Japanese medical students in PBL are inactive. Therefore, the students’ interactions during PBL may differ from those in other English-speaking countries [22]. This could influence the response to the items of the help-seeking and peer-learning subscales, as it is considered that there is a social-interaction pattern within the learning environment that is unique to Japanese medical trainees [18,23].

### 4.4. Extrapolation Inference

Due to its limited coherence with theory, the learning-strategies section is not applicable for usage, as the validity evidence in both the literature and the present study for the section was insufficient. On the contrary, the motivation section was well-validated from both perspectives.

There are limitations to arguing the implications of the J-MSLQ-PBL. First, as discussed, there would be an issue of face validity due to the insufficient translation procedure; thus, the team translation model needs to be applied to consider the cultural influence on the translation process. In addition, integrating qualitative evidence, such as in-depth interviews to explore the students’ perceptions of item descriptions, could help improve the face validity.

## 5. Conclusions

This study examined validity evidence for the Japanese version of the MSLQ adapted to the PBL context. The internal structure of the motivation section of the J-MSLQ-PBL was consistent with the theory of self-regulated learning; however, the learning-strategies section did not align with the original structure of the MSLQ. Thus, only the motivation section of the J-MSLQ-PBL can be implemented to measure competency elements in self-regulated learning in Japan. Additional work is needed to clarify the reason for the discrepancy in the learning-strategies section between the J-MSLQ-PBL and the MSLQ.

## Figures and Tables

**Figure 1 children-10-00154-f001:**
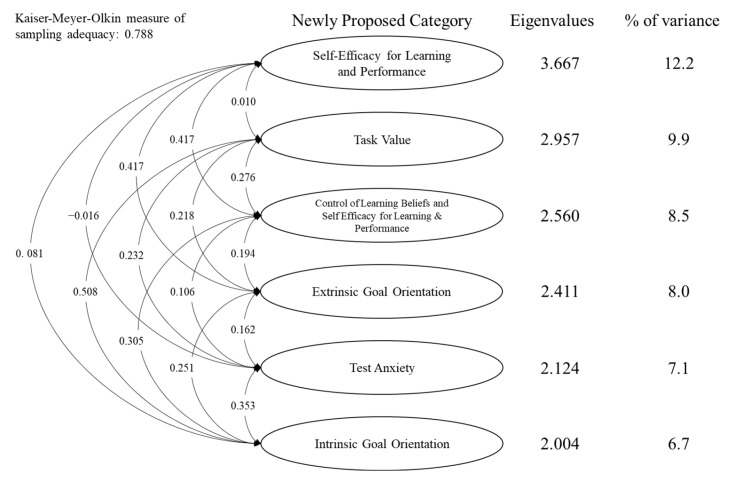
Factor analysis of the Motivation Scale.

**Figure 2 children-10-00154-f002:**
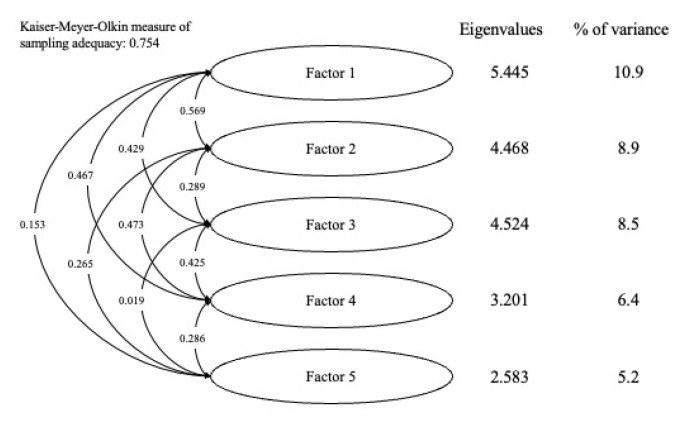
Factor analysis of the Learning Strategies Scale.

**Table 1 children-10-00154-t001:** Descriptive statistics of the Motivation Scale.

No.	Items	Mean	SD	Median	25th–75th Percentile
Q1	In a class like this, I prefer course material that really challenges me so I can learn new things.	3.393	1.618	3	2–4.25
Q2	If I study in appropriate ways, then I will be able to learn the material in the medical course.	5.134	1.319	5	4–6
Q3	When I take a test, I think about how poorly I am doing compared to other students.	4.330	1.862	4.5	3–6
Q4	I think I will be able to use what I learn in the medical course in other courses.	5.089	1.545	5	4–6
Q5	I believe I will receive an excellent grade in this class.	2.991	1.580	3	2–4
Q6	I’m certain I can understand the most difficult material presented in the readings for the medical course.	2.732	1.530	2	2–3
Q7	Getting a good grade in this class is the most satisfying thing for me right now.	3.161	1.669	3	2–4
Q8	When I take a test, I think about items on other parts of the test I can’t answer.	4.179	1.715	5	3–5
Q9	It is my own fault if I don’t learn the material in the medical course.	4.277	1.623	4	3–5
Q10	It is important for me to learn the course material in the medical class.	4.714	1.290	5	4–5.25
Q11	The most important thing for me right now is improving my overall grade point average, so my main concern in this class is getting a good grade.	3.482	1.524	3	2–4.25
Q12	I’m confident I can learn the basic concepts taught in the medical course.	4.036	1.445	4	3–5
Q13	If I can, I want to get better grades in this class than most of the other students.	4.304	1.898	4	3–6
Q14	When I take tests, I think of the consequences of failing.	3.491	1.959	3	2–5
Q15	I’m confident I can understand the most complex material presented by the instructor in this course.	2.661	1.516	2	1.75–4
Q16	In a class like this, I prefer course material that arouses my curiosity, even if it is difficult to learn.	4.545	1.638	5	3–6
Q17	I am very interested in the content area of the medical course.	5.071	1.400	5	4–6
Q18	If I try hard enough, then I will understand the course material.	4.991	1.424	5	4–6
Q19	I have an uneasy upset feeling when I take an exam.	3.786	1.747	3.5	3–5
Q20	I’m confident I can do an excellent job on the assignments and tests in the medical course.	3.054	1.512	3	2–4
Q21	I expect to do well in this course.	3.188	1.685	3	2–4
Q22	The most satisfying thing for me in the medical course is trying to understand the content as thoroughly as possible.	3.982	1.483	4	3–5
Q23	I think the material for the PBL course is useful for me to learn.	5.420	1.235	6	5–6
Q24	When I have the opportunity in this class, I choose course assignments that I can learn from, even if they don’t guarantee a good grade.	3.804	1.734	4	2–5
Q25	If I don’t understand the course material, it is because I didn’t try hard enough.	4.000	1.458	4	3–5
Q26	I like the subject matter of the medical course.	4.607	1.311	5	4–6
Q27	Understanding the subject matter of the medical course is very important to me.	4.920	1.246	5	4–6
Q28	I feel my heart beating fast when I take an exam.	3.911	1.920	4	2–5.25
Q29	I’m certain I can master the skills being taught in this class.	3.411	1.480	3	2–4
Q30	I want to do well in this class because it is important to show my ability to my family, friends, employer, or others.	3.089	1.696	3	2–4
Q31	Considering the difficulty of the medical course, the teacher, and my skills, I think I will do well in this course.	2.893	1.538	3	2–4

**Table 2 children-10-00154-t002:** Descriptive statistics of the Learning Strategies Scale.

No.	Items	Mean	SD	Median	25th–75th Percentile
Q32	When I study the readings for the medical course, I outline the material to help me organize my thoughts.	4.009	1.690	4	3–5
Q33	During class time, I often miss important points because I’m thinking of other things.	3.366	1.483	3	2–4
Q34	When studying for the medical course, I often try to explain the material to a classmate or friend.	3.759	1.607	4	2–5
Q35	I usually study in a place where I can concentrate on my course work.	4.821	1.584	5	4–6
Q36	When reading for the medical course, I make up questions to help focus my reading.	3.188	1.492	3	2–4
Q37	I often feel so lazy or bored when I study for this PBL course that I quit before I finish what I planned to do.	4.518	1.548	5	4–6
Q38	I often find myself questioning things I hear or read in the medical course to decide if I find them convincing.	4.179	1.472	4	3–5
Q39	When I study for this PBL course, I practice saying the material to myself over and over.	2.955	1.491	3	2–4
Q40	Even if I have trouble learning the material in this class, I try to do the work on my own, without help from anyone.	4.902	1.682	5	4–6
Q41	When I become confused about something I’m reading for this PBL course, I go back and try to figure it out.	4.679	1.543	5	4–6
Q42	When I study for the medical course, I go through the readings and my class notes and try to find the most important ideas.	4.518	1.513	5	4–6
Q43	I make good use of my study time for the medical course.	4.304	1.476	4	3–5
Q44	If the medical course readings are difficult to understand, I change the way I read the material.	4.509	1.427	5	4–5
Q45	I try to work with other students from this PBL course to complete the course assignments.	5.259	1.457	6	4–6
Q46	When studying for the medical course, I read my class notes and the course readings over and over again.	4.750	1.574	5	4–6
Q47	When a theory, interpretation, or conclusion is presented in class or in the readings, I try to decide if there is good supporting evidence.	4.089	1.534	4	3–5
Q48	I work hard to do well in this PBL course, even if I don’t like what we are doing.	4.813	1.353	5	4–6
Q49	I make simple charts, diagrams, or tables to help me organize the course material.	3.821	1.667	4	2.75–5
Q50	When studying for the medical course, I often set aside time to discuss the course material with a group of students from the class.	4.580	1.493	5	4–6
Q51	I treat the course material as a starting point and try to develop my own ideas about it.	4.196	1.334	4	3–5
Q52	I find it hard to stick to a study schedule.	3.366	1.577	3	2–5
Q53	When I study for this PBL course, I pull together information from different sources, such as lectures, readings, and discussions.	4.643	1.328	5	4–6
Q54	Before I study new course material thoroughly, I often skim it to see how it is organized.	4.402	1.557	5	3–5.25
Q55	I ask myself questions to make sure I understand the material I have been studying in this PBL course.	4.054	1.426	4	3–5
Q56	I try to change the way I study in order to fit the course requirements and the instructor’s teaching style.	4.259	1.354	4	3–5
Q57	I often find that I have been reading for this PBL course but don’t know what it was all about.	3.929	1.609	4	3–5
Q58	I ask the instructor to clarify concepts I don’t understand well.	3.295	1.540	3	2–4
Q59	I memorize key words to remind me of important concepts in this class.	4.438	1.406	5	3–5
Q60	When course work is difficult, I either give up or only study the easy parts.	3.830	1.542	4	3–5
Q61	I try to think through a topic and decide what I am supposed to learn from it rather than just reading it over when studying for the medical course.	3.964	1.506	4	3–5
Q62	I try to relate ideas in this subject to those in other courses whenever possible.	4.777	1.380	5	4–6
Q63	When I study for medical course, I go over my class notes and make an outline of important concepts.	4.188	1.630	4	3–5.25
Q64	When reading for this this PBL course, I try to relate the material to what I already know.	4.964	1.420	5	4–6
Q65	I have a regular place set aside for studying.	4.679	1.731	5	3–6
Q66	I try to play around with ideas of my own related to what I am learning in the medical course.	4.384	1.377	4	3–5
Q67	When I study for the medical course, I write brief summaries of the main ideas from the readings and my class notes.	3.920	1.730	4	3–5
Q68	When I can’t understand the material in the medical course, I ask another student in this class for help.	5.482	1.315	6	5–6.25
Q69	I try to understand the material in this PBL course by making connections between the readings and the concepts from the lectures.	5.089	1.305	5	5–6
Q70	I make sure that I keep up with the weekly readings and assignments for the medical course.	3.696	1.547	4	3–5
Q71	Whenever I read or hear an assertion or conclusion in this class, I think about possible alternatives.	3.866	1.417	4	3–5
Q72	I make lists of important items for the medical course and memorize the lists.	3.304	1.765	3	2–5
Q73	I attend this class regularly.	4.536	1.869	5	3–6
Q74	Even when the course materials are dull and uninteresting, I manage to keep working until I finish.	4.402	1.479	5	3–5
Q75	I try to identify students in this class to ask for help if necessary.	4.384	1.767	5	3–6
Q76	When studying for the medical course, I try to determine which concepts I don’t understand well.	4.625	1.396	5	4–5
Q77	I often find that I don’t spend very much time on the medical course because of other activities.	3.384	1.590	3	2–4
Q78	When I study for this PBL course, I set goals for myself in order to direct my activities in each study period.	3.875	1.440	4	3–5
Q79	If I get confused taking notes in class, I make sure I sort it out afterward.	3.732	1.644	4	2.75–5
Q80	I rarely find time to review my notes or readings before an exam.	4.786	1.608	5	4–6
Q81	I try to apply ideas from the course readings in other class activities such as lectures and discussions.	4.545	1.451	5	4–5

Note: Q33, 37, 40, 52, 57, 60, 77, and 80 are the inverted scales and were appropriately adjusted for the analysis.

**Table 3 children-10-00154-t003:** Factor loadings and variance explained from an exploratory factor analysis of the Motivation Scale.

No.	Self-Efficacy for Learning and Performance	Task Value	Control of Learning Beliefs and Self-Efficacy for Learning and Performance	Extrinsic Goal Orientation	Test Anxiety	Intrinsic Goal Orientation
	Cronbach’s α	0.875				
Q15	0.908	0.075	–0.113	0.001	–0.013	0.147
Q6	0.873	0.101	–0.080	–0.078	0.090	0.066
Q31	0.679	–0.011	0.131	0.185	0.106	–0.096
Q20	0.620	0.146	0.246	0.006	–0.081	–0.159
Q12	0.395	0.150	0.183	–0.032	–0.015	0.173
	Cronbach’s α	0.805				
Q17	0.051	0.827	0.069	0.048	–0.177	–0.091
Q26	0.273	0.817	–0.078	–0.029	–0.074	–0.007
Q27	0.209	0.716	–0.052	–0.020	0.210	–0.025
Q23	–0.263	0.513	0.142	0.021	0.009	0.133
Q9	–0.129	0.240	–0.091	0.101	0.239	0.159
	Cronbach’s α	0.800				
Q21	0.221	–0.073	0.729	0.253	–0.036	–0.313
Q5	0.344	–0.298	0.610	0.151	–0.031	0.069
Q18	–0.042	0.260	0.570	–0.050	–0.025	0.136
Q29	0.313	–0.046	0.511	0.075	0.002	–0.039
Q2	0.053	0.096	0.436	–0.104	0.197	0.104
Q4	–0.068	0.251	0.317	–0.225	0.102	0.160
	Cronbach’s α	0.792				
Q7	0.117	–0.165	0.084	0.730	–0.053	0.170
Q13	–0.074	0.298	0.180	0.653	–0.007	–0.246
Q30	–0.003	–0.285	0.181	0.621	0.138	0.158
Q11	0.103	0.148	–0.315	0.599	0.165	0.084
Q22	0.083	0.258	–0.099	0.426	–0.048	0.256
	Cronbach’s α	0.720				
Q19	0.126	0.038	0.025	0.022	0.873	–0.261
Q28	–0.095	0.130	0.098	0.150	0.844	–0.344
Q8	0.212	–0.059	0.081	–0.034	0.480	0.107
Q14	–0.122	–0.194	–0.042	0.017	0.452	0.194
	Cronbach’s α	0.638				
Q1	0.307	–0.064	0.035	0.006	–0.128	0.621
Q25	–0.037	–0.001	–0.114	0.037	–0.105	0.595
Q24	0.044	–0.041	0.113	–0.034	0.089	0.523
Q10	–0.158	0.179	0.024	0.209	–0.060	0.455
Q16	–0.018	0.197	0.270	0.075	–0.159	0.393
Eigenvalues	3.667	2.957	2.560	2.411	2.124	2.004
Percentage of total variance	0.122	0.099	0.085	0.080	0.071	0.067

**Table 4 children-10-00154-t004:** Factor loadings and variance explained from an exploratory factor analysis of the Learning Strategies Scale.

No.	Factor 1	Factor 2	Factor 3	Factor 4	Factor 5
	Cronbach’s α	0.805			
Q42	0.715	–0.101	–0.128	0.167	–0.144
Q74	0.684	–0.148	–0.017	0.043	–0.034
Q73	0.674	–0.298	0.156	–0.177	0.074
Q48	0.654	0.093	0.011	–0.238	0.014
Q76	0.620	0.105	–0.051	–0.117	–0.039
Q41	0.574	0.190	–0.195	0.022	–0.134
Q67	0.565	–0.275	0.209	0.108	0.306
Q51	0.548	0.057	0.098	0.065	–0.174
Q59	0.526	0.264	–0.072	–0.088	–0.120
Q52	–0.482	0.009	–0.135	0.475	0.399
Q32	0.441	–0.055	–0.093	0.315	0.125
Q81	0.400	0.146	0.002	0.177	0.023
Q35	0.392	0.094	–0.328	0.359	–0.048
Q62	0.365	0.183	0.137	0.135	0.038
Q47	0.295	0.090	0.213	0.081	–0.138
	Cronbach’s α	0.824			
Q45	–0.196	0.963	–0.109	–0.153	–0.266
Q68	0.046	0.835	0.034	–0.290	–0.030
Q50	–0.192	0.650	0.284	0.182	–0.220
Q46	0.122	0.462	0.021	0.145	0.056
Q64	0.219	0.442	–0.013	0.160	0.014
Q44	0.121	0.423	0.099	0.026	–0.067
Q43	0.027	0.409	0.189	0.187	0.022
Q69	0.348	0.372	0.097	0.028	0.086
Q66	0.080	0.310	0.208	0.209	0.020
	Cronbach’s α	0.650			
Q78	–0.049	0.015	0.683	–0.097	0.144
Q57	–0.101	0.028	–0.603	0.379	0.260
Q72	–0.201	–0.088	0.589	0.330	0.161
Q55	–0.034	0.223	0.573	0.071	–0.120
Q58	–0.070	–0.031	0.559	0.042	–0.199
Q71	–0.118	0.302	0.466	–0.029	–0.019
Q79	0.122	−0.023	0.453	0.211	0.391
Q75	0.105	0.288	0.446	–0.231	0.056
Q39	0.161	–0.198	0.399	0.203	–0.272
Q53	0.201	0.083	0.383	0.005	0.080
Q54	0.163	0.093	0.324	0.063	0.112
	Cronbach’s α	0.496			
Q36	–0.022	–0.291	0.084	0.845	–0.337
Q61	0.204	0.170	–0.083	0.459	–0.309
Q70	–0.031	–0.023	0.216	0.446	0.054
Q34	–0.138	0.301	0.228	0.324	–0.010
Q56	0.086	0.225	0.197	0.293	0.002
Q38	0.080	0.108	0.155	0.281	0.045
Q65	0.120	0.167	–0.028	0.266	0.034
	Cronbach’s α	0.685			
Q37	–0.060	0.178	–0.087	–0.227	0.661
Q77	–0.096	–0.137	0.074	–0.219	0.624
Q33	0.228	–0.313	–0.115	0.059	0.559
Q60	–0.158	–0.175	0.146	0.037	0.488
Q63	0.223	0.165	0.255	0.062	0.366
Q80	0.090	0.300	–0.283	–0.111	0.318
Eigenvalues	5.445	4.468	4.254	3.201	2.583
Percentage of total variance	0.109	0.089	0.085	0.064	0.052

## Data Availability

Not applicable.

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
