# Peer review of "Adapting the Motivated Strategies for Learning Questionnaire to the Japanese Problem-Based Learning Context: A Validation Study"

_children, 2023, doi:10.3390/children10010154_

Round 1
Reviewer 1 Report
This was a secondary analysis study using a database, which was initially collected to examine the impact of self-regulated learning during the problem-based learning (PBL) course at Jichi Medical University, Japan, on medical students’ professional identity formation. Of all 124 third-year medical students invited to participate in this study, 112 agreed.Author Response
Thank you very much for your supportive comment.
Reviewer 2 Report
The authors of the study demonstrate that it is not easy to transfer theoretical studies easily into practice. There must be a suitability study that takes into account the culture of each people. This finding makes the work useful.
Author Response
Thank you so much for your supportive comment.
Reviewer 3 Report
In this article authors aimed to validate the translated Japanese version of the Motivated Strategies for Learning Questionnaire in the context of Problem-based Learning (J-MSLQ-PBL). The questionnaire employs a seven-point Likert-type scale with 81 items and is categorized into two sections: Motivation and learning strategies. Exploratory factor analysis (EFA) was conducted using Promax rotation to examine the factor structure of the scale using the collected data from 112 Japanese medical students. Factor extraction was based on scree plot investigation, and an item was accepted when the factor loading was ≥0.40. In the motivation section, the extracted factors from the EFA were well aligned with the subscales of the original MSLQ, including “Self-Efficacy for Learning and Performance,” “Task Value,” “Self-Efficacy for 23
Learning and Performance,” “Test Anxiety,” “Extrinsic Goal Orientation,” and “Intrinsic Goal Orientation.” In the learning strategies, the extracted factors poorly matched the structure of the original subscales. This discrepancy autors explained by insufficient translation, the limited sample size from a single medical school, or cross-cultural differences in learning strategies between Western and Japanese medical students.
Author Response
Thank you for your comment.
Since we could not find any specific comments for the revision, we have corrected our paper in response to another reviewer's comment.
Reviewer 4 Report
Dear authors,
You have done a great job. However, the introduction and conclusion sections are not appropriately elaborated on. Please make modifications and add the required changes to the revised manuscript.
Kind regards,
Author Response
Thank you for your valuable comments.
In response to your suggestions, we elaborated on the descriptions of the Introduction(Line 39-44, Line 57-68) and Conclusions (L235-238).
Round 2
Reviewer 4 Report
Dear authors
Thank you very much indeed for considering the required modifications in the revised manuscript.
Kind regards,